# DeepTOP: Deep Threshold-Optimal Policy for MDPs and RMABs

**Khaled Nakhleh**        **I-Hong Hou**

Electrical and Computer Engineering Department
Texas A&M University
College Station, TX
{khaled.jamal, ihou}@tamu.edu

## Abstract

We consider the problem of learning the optimal threshold policy for control problems. Threshold policies make control decisions by evaluating whether an element of the system state exceeds a certain threshold, whose value is determined by other elements of the system state. By leveraging the monotone property of threshold policies, we prove that their policy gradients have a surprisingly simple expression. We use this simple expression to build an off-policy actor-critic algorithm for learning the optimal threshold policy. Simulation results show that our policy significantly outperforms other reinforcement learning algorithms due to its ability to exploit the monotone property.

In addition, we show that the Whittle index, a powerful tool for restless multi-armed bandit problems, is equivalent to the optimal threshold policy for an alternative problem. This observation leads to a simple algorithm that finds the Whittle index by learning the optimal threshold policy in the alternative problem. Simulation results show that our algorithm learns the Whittle index much faster than several recent studies that learn the Whittle index through indirect means.

## 1   Introduction

This paper considers a class of control policies, called *threshold policies*, that naturally arise in many practical problems. For example, a smart home server may only turn on the air conditioner when the room temperature exceeds a certain threshold, and a central bank may only raise the interest rate when inflation exceeds a certain threshold. For such problems, finding the optimal control policies can be reduced to finding the appropriate thresholds given other factors of the system, such as the number of people in the room in the smart home server scenario or the unemployment rate and the current interest rate in the central bank scenario.

An important feature of threshold policies is that their actions are monotone. For example, if a smart home server would turn on the air conditioner at a certain temperature, then, all other factors being equal, the server would also turn on the air conditioner when the temperature is even higher. By leveraging this monotone property, an algorithm aiming to learn the optimal threshold can potentially be much more efficient than generic reinforcement learning algorithms seeking to learn the optimal action at different points of temperature separately. In order to design an efficient algorithm for learning the optimal threshold policy, we first formally define a class of Markov decision processes (MDPs) that admit threshold policies and its objective function. The optimal threshold policy is then the one that maximizes the objective function. However, the objective function involves an integral over a continuous range, which makes it infeasible to directly apply standard tools, such as backward-propagation in neural networks, to perform gradient updates.

36th Conference on Neural Information Processing Systems (NeurIPS 2022).

Surprisingly, we show that, by leveraging the monotone property of threshold policies, the gradient of the objective function has a very simple expression. Built upon this expression, we propose Deep Threshold-Optimal Policy (DeepTOP), a model-free actor-critic deep reinforcement learning algorithm that finds the optimal threshold policies. We evaluate the performance of DeepTOP by considering three practical problems, an electric vehicle (EV) charging problem that determines whether to charge an EV in the face of unknown fluctuations of electricity price, an inventory management problem that determines whether to order for goods in the face of unknown seasonal demands, and a make-to-stock problem for servicing jobs with different sizes. For all problems, DeepTOP significantly outperforms other state-of-the-art deep reinforcement learning algorithms due to its ability to exploit the monotone property.

We also study the notoriously hard restless multi-armed bandit (RMAB) problem. We show that the Whittle index policy, a powerful tool for RMABs, can be viewed as an optimal threshold policy for an alternative problem. Based on this observation, we define an objective function for the alternative problem, of which the Whittle index is the maximizer. We again show that the gradient of the objective function has a simple expression. This simple expression allows us to extend DeepTOP for the learning of the Whittle index. We compare this DeepTOP extension to three recently proposed algorithms that seek to learn the optimal index policies through other indirect properties. Simulation results show that the DeepTOP extension learns much faster because it directly finds the optimal threshold policy.

The rest of the paper is organized as follows. Section 2 defines the MDP setting and threshold policies. We present the DeepTOP algorithm for MDP in Section 3. We then discuss how the Whittle index policy for RMABs can be viewed as a threshold policy in Section 4 and develop a DeepTOP extension for learning it in Section 5. We show DeepTOP's performance results for MDPs and RMABs in Section 6, and give related works in Section 7 before concluding.

## 2 Threshold Policies for MDPs

Consider an agent controlling a stochastic environment $\mathcal{E}$ described as an MDP $\mathcal{E} = (\mathcal{S}, \mathcal{A}, \mathcal{R}, \mathcal{P}, \gamma)$, with state space $\mathcal{S}$, binary action space $\mathcal{A} := \{0, 1\}$, reward function $\mathcal{R} : \mathcal{S} \times \mathcal{A} \to \Omega$, transition dynamics $\mathcal{P} : \mathcal{S} \times \mathcal{A} \times \mathcal{S} \to \mathbb{R}$, and discount factor $\gamma \in [0, 1)$, where $\mathbb{R}$ is the set of real numbers and $\Omega$ is the set of random variables. At each timestep $t$, the agent picks an action $a_t \in \mathcal{A}$ for the current state $s_t$. The state $s_t \in \mathcal{S} = \mathbb{R} \times \mathcal{V}$ has two components: a scalar state $\lambda_t \in \mathbb{R}$, and a vector state $v_t \in \mathcal{V}$, where $\mathcal{V}$ is a discrete set of vectors. We assume the environment state is fully observable. Given the state-action pair $(s_t, a_t)$, the MDP generates a reward $r_t$ following the unknown random variable $\mathcal{R}(s_t, a_t)$, and a random next state $s_{t+1} = (\lambda_{t+1}, v_{t+1})$ following the unknown distribution $\mathcal{P}$. We use $\bar{r}(\lambda, v, a) := E[\mathcal{R}((\lambda, v), a)]$ to denote the unknown expected one-step reward that can be obtained for the state-action pair $(\lambda, v, a)$.

A threshold policy is one that defines a threshold function $\mu : \mathcal{V} \to \mathbb{R}$ mapping each vector state to a real number. The policy then deterministically picks $a_t = \mathbb{1}(\mu(v_t) > \lambda_t)$, where $\mathbb{1}(\cdot)$ is the indicator function. There are many applications where it is natural to consider threshold policies and we discuss some of them below.

**Example 1.** *Consider the problem of charging electric vehicles (EV). When an EV arrives at a charging station, it specifies its demands for charge and a deadline upon which it will leave the station. The electricity price changes over time following some random process. The goal of the operator is to fulfill the EV's requirement with minimum cost. In this problem, we can model the system by letting the scalar state $\lambda_t$ be the current electricity price and the vector state $v_t$ be the remaining charge and time to deadline of the EV. For this problem, it is natural to consider a threshold policy that defines a threshold $\mu(v_t)$ as the highest price the operator is willing to pay to charge the EV under vector state $v_t$. The operator only charges the vehicle, i.e., chooses $a_t = 1$, if $\lambda_t < \mu(v_t)$.*

**Example 2.** *Consider the problem of warehouse management. A warehouse stores goods waiting to be sold. When the number of stored goods exceeds the demand, then there is a holding cost for each unsold good. On the other hand, if the number of stored goods is insufficient to fulfill the demand, then there is a cost of lost sales. The goal of the manager is to decide when to place orders so as to minimize the total cost. In this problem we can let the scalar state $\lambda_t$ be the current inventory and let the vector state $v_t$ be the vector of all factors, such as upcoming holidays, that can influence future demands. It is natural to consider a threshold policy where the manager only places a new order if the current inventory $\lambda_t$ falls below a threshold $\mu(v_t)$ based on the current vector state $v_t$.*

**Example 3.** *Consider a smart home server that controls the air conditioner. Let $\lambda_t$ be −(current temperature) and $v_t$ be the time of the day and the number of people in the house. The server should turn on the air conditioner only if the temperature exceeds some threshold determined by $v_t$, or, equivalently, $\lambda_t < \mu(v_t)$.*

Given a threshold policy with threshold function $\mu(\cdot)$, we can define the corresponding action-value function by $Q_\mu(\lambda, v, a)$. Let $\rho_\mu(\lambda', v', \lambda, v)$ be the discounted state distribution when the initial state is $(\lambda, v)$ under the threshold policy to a visited state $(\lambda', v')$. When the initial state is $(\lambda, v)$, the expected discounted reward under the policy is

$$Q_\mu\big(\lambda, v, \mathbb{1}(\mu(v) > \lambda)\big) = \sum_{v' \in \mathcal{V}} \int_{\lambda'=-M}^{\lambda'=+M} \rho_\mu(\lambda', v', \lambda, v)\bar{r}(\lambda', v', \mathbb{1}(\mu(v') > \lambda')). \tag{1}$$

Let $M$ be a sufficiently large constant such that $\lambda_t \in [-M, +M]$ for all $t$. Our goal is to learn the optimal threshold function $\mu^\phi(v)$ parametrized by a vector $\phi$ that maximizes the objective function

$$K(\mu^\phi) := \int_{\lambda=-M}^{\lambda=+M} \sum_{v \in \mathcal{V}} Q_{\mu^\phi}\big(\lambda, v, \mathbb{1}(\mu^\phi(v) > \lambda)\big)d\lambda. \tag{2}$$

## 3 Deep Threshold Optimal Policy for MDPs

In this section, we present a deep threshold optimal policy (DeepTOP) for MDPs that finds the optimal $\phi$ for maximizing $K(\mu^\phi)$.

### 3.1 Threshold Policy Gradient Theorem for MDPs

In order to design DeepTOP, we first study the gradient $\nabla_\phi K(\mu^\phi)$. At first glance, computing $\nabla_\phi K(\mu^\phi)$ looks intractable since it involves an integral over $\lambda \in [-M, +M]$. However, we establish the following threshold policy gradient theorem that shows the surprising result that $\nabla_\phi K(\mu^\phi)$ has a simple expression.

**Theorem 1.** *Given the parameter vector $\phi$, let $\bar{\rho}(\lambda, v)$ be the discounted state distribution when the initial state is chosen uniformly at random under the threshold policy. If all vector states $v \in \mathcal{V}$ have distinct values of $\mu^\phi(v)$, then,*

$$\nabla_\phi K(\mu^\phi) = 2M|\mathcal{V}| \sum_{v \in \mathcal{V}} \bar{\rho}(\mu^\phi(v), v)\big(Q_{\mu^\phi}(\mu^\phi(v), v, 1) - Q_{\mu^\phi}(\mu^\phi(v), v, 0)\big)\nabla_\phi\mu^\phi(v). \tag{3}$$

*Proof.* Let $\bar{\rho}_t(\lambda, v)$ be the distribution that the state at time $t$ is $(\lambda, v)$ when the initial state is chosen uniformly at random. Clearly, we have $\bar{\rho}(\lambda, v) = \sum_{t=1}^{\infty} \gamma^{t-1}\bar{\rho}_t(\lambda, v)$. Given $\phi$, we number all states in $\mathcal{V}$ such that $\mu^\phi(v^1) > \mu^\phi(v^2) > \dots$. Let $\mathbb{M}^0 = +M$, $\mathbb{M}^n = \mu^\phi(v^n)$, for all $1 \le n \le |\mathcal{V}|$, and $\mathbb{M}^{|\mathcal{V}|+1} = -M$. Also, let $\mathbb{V}^n$ be the subset of states $\{v|\mu^\phi(v) > \mathbb{M}^n\} = \{v^1, v^2, \dots, v^{n-1}\}$. Now, consider the interval $(\mathbb{M}^{n+1}, \mathbb{M}^n)$ for some $n$. Notice that, for all $\lambda \in (\mathbb{M}^{n+1}, \mathbb{M}^n)$, $\mathbb{1}(\mu^\phi(v) > \lambda) = 1$ if and only if $v \in \mathbb{V}^{n+1}$. In other words, for any vector state $v$, the threshold policy would take the same action under all $\lambda \in (\mathbb{M}^{n+1}, \mathbb{M}^n)$, and we use $\pi^{n+1}(v)$ to denote this action. We then have

$$\nabla_\phi K(\mu^\phi) = \nabla_\phi \int_{\lambda=-M}^{\lambda=+M} \sum_{v \in \mathcal{V}} Q_{\mu^\phi}(\lambda, v, \mathbb{1}(\mu^\phi(v) > \lambda))d\lambda = \sum_{v \in \mathcal{V}} \nabla_\phi \int_{\lambda=-M}^{\lambda=+M} Q_{\mu^\phi}(\lambda, v, \mathbb{1}(\mu^\phi(v) > \lambda))d\lambda$$

$$= \sum_{v \in \mathcal{V}} \sum_{n=0}^{|\mathcal{V}|} \nabla_\phi \int_{\lambda=\mathbb{M}^{n+1}}^{\lambda=\mathbb{M}^n} Q_{\mu^\phi}(\lambda, v, \pi^{n+1}(v))d\lambda$$

$$= \sum_{v \in \mathcal{V}} \sum_{n=0}^{|\mathcal{V}|} \bigg(Q_{\mu^\phi}(\mathbb{M}^n, v, \pi^{n+1}(v))\nabla_\phi\mathbb{M}^n - Q_{\mu^\phi}(\mathbb{M}^{n+1}, v, \pi^{n+1}(v))\nabla_\phi\mathbb{M}^{n+1} + \int_{\lambda=\mathbb{M}^{n+1}}^{\lambda=\mathbb{M}^n} \nabla_\phi Q_{\mu^\phi}(\lambda, v, \pi^{n+1}(v))d\lambda\bigg),$$
$$\tag{4}$$

where the summation-integration swap in the first equation follows the Fubini-Tonelli theorem and the last step follows the Leibniz integral rule. We simplify the first two terms in the last step by

$$\sum_{v \in \mathcal{V}} \sum_{n=0}^{|\mathcal{V}|} \left( Q_{\mu^\phi}(\mathbb{M}^n, v, \pi^{n+1}(v)) \nabla_\phi \mathbb{M}^n - Q_{\mu^\phi}(\mathbb{M}^{n+1}, v, \pi^{n+1}(v)) \nabla_\phi \mathbb{M}^{n+1} \right)$$

$$= \sum_{v \in \mathcal{V}} \sum_{n=1}^{|\mathcal{V}|} \left( Q_{\mu^\phi}(\mu^\phi(v^n), v, \mathbb{1}(v \in \mathbb{V}^{n+1})) - Q_{\mu^\phi}(\mu^\phi(v^n), v, \mathbb{1}(v \in \mathbb{V}^n)) \right) \nabla_\phi \mu^\phi(v^n)$$

$$= 2M|\mathcal{V}| \sum_{v \in \mathcal{V}} \bar{\rho}_1(\mu^\phi(v), v) \left( Q_{\mu^\phi}(\mu^\phi(v), v, 1) - Q_{\mu^\phi}(\mu^\phi(v), v, 0) \right) \nabla_\phi \mu^\phi(v). \tag{5}$$

Next, we expand the last term in (4). Note that $Q_{\mu^\phi}(\lambda, v, a) = \bar{r}(\lambda, v, a) + \gamma \int_{\lambda'=-M}^{\lambda'=+M} \sum_{v'} p(\lambda', v'|\lambda, v, a) Q_{\mu^\phi}(\lambda', v', \mathbb{1}(\mu^\phi(v') > \lambda')) d\lambda'$, where $p(\cdot|\cdot)$ is the transition probability. Hence, $\nabla_\phi Q_{\mu^\phi}(\lambda, v, a) = \gamma \nabla_\phi \int_{\lambda'=-M}^{\lambda'=+M} \sum_{v'} p(\lambda', v'|\lambda, v, a) Q_{\mu^\phi}(\lambda', v', \mathbb{1}(\mu^\phi(v') > \lambda')) d\lambda'$. Using the same techniques in (4) and (5), we have

$$\sum_{v \in \mathcal{V}} \sum_{n=0}^{|\mathcal{V}|} \int_{\lambda=\mathbb{M}^{n+1}}^{\lambda=\mathbb{M}^n} \nabla_\phi Q_{\mu^\phi}(\lambda, v, \pi^{n+1}(v)) d\lambda = \sum_{v \in \mathcal{V}} \int_{\lambda=-M}^{\lambda=+M} \nabla_\phi Q_{\mu^\phi}(\lambda, v, \mathbb{1}(\mu^\phi(v) > \lambda)) d\lambda$$

$$= \gamma \sum_{v \in \mathcal{V}} \int_{\lambda=-M}^{\lambda=+M} \left( \nabla_\phi \int_{\lambda'=-M}^{\lambda'=+M} \sum_{v' \in \mathcal{V}} p(\lambda', v'|\lambda, v, \mathbb{1}(\mu^\phi(v) > \lambda)) Q_{\mu^\phi}(\lambda', v', \mathbb{1}(\mu^\phi(v') > \lambda')) d\lambda' \right) d\lambda$$

$$= 2M|\mathcal{V}| \sum_{v \in \mathcal{V}} \gamma \bar{\rho}_2(\mu^\phi(v), v) \left( Q_{\mu^\phi}(\mu^\phi(v), v, 1) - Q_{\mu^\phi}(\mu^\phi(v), v, 0) \right) \nabla_\phi \mu^\phi(v)$$

$$+ \gamma \sum_{v \in \mathcal{V}} \int_{\lambda=-M}^{\lambda=+M} \left( \sum_{v' \in \mathcal{V}} \int_{\lambda'=-M}^{\lambda'=+M} \nabla_\phi(p(\lambda', v'|\lambda, v, \mathbb{1}(\mu^\phi(v) > \lambda)) Q_{\mu^\phi}(\lambda, v, \mathbb{1}(\mu^\phi(v') > \lambda'))) d\lambda' \right) d\lambda.$$

In the above equation, expanding the last term in time establishes (3). □

### 3.2 DeepTOP Algorithm Design for MDPs

Motivated by Theorem 1, we now present DeepTOP-MDP, a model-free, actor-critic Deep RL algorithm. DeepTOP-MDP maintains an actor network with parameters $\phi$ that learns a threshold function $\mu^\phi(v)$, and a critic network with parameters $\theta$ that learns an action-value function $Q^\theta(\lambda, v, a)$. DeepTOP-MDP also maintains a target critic network with parameters $\theta'$ that is updated slower than the critic parameters $\theta$. The purpose of the target critic network is to improve the learning stability as demonstrated in [8, 19]. The objective of the critic network is to find $\theta$ that minimizes the loss function

$$\mathcal{L}(\theta) := \mathbb{E}_{s_t, a_t, r_t, s_{t+1}} \left[ \left( Q^\theta(\lambda_t, v_t, a_t) - r_t - \gamma \max_{a' \in \mathcal{A}} Q^{\theta'}(\lambda_{t+1}, v_{t+1}, a') \right)^2 \right], \tag{6}$$

where $(s_t, a_t, r_t, s_{t+1})$ is sampled under some policy with $s_t = (\lambda_t, v_t)$. The objective of the actor network is to find $\phi$ that maximizes $\int_{\lambda=-M}^{\lambda=+M} \sum_{v \in \mathcal{V}} Q_{\mu^\phi}^\theta \left( \lambda, v, \mathbb{1}(\mu^\phi(v) > \lambda) \right) d\lambda$. In each timestep $t$, the environment $\mathcal{E}$ provides a state $s_t$ to the agent. We set an exploration parameter $\epsilon_t \in [0, 1)$ that takes a random action with probability $\epsilon_t$. Otherwise, DeepTOP-MDP calculates $\mu^\phi(v_t)$ based on $v_t$, and chooses $a_t = \mathbb{1}(\mu^\phi(v_t) > \lambda_t)$. $\mathcal{E}$ generates a reward $r_t$ and a next state $s_{t+1}$. A replay memory denoted by $\mathcal{M}$ then stores the transition $\{s_t, a_t, r_t, s_{t+1}\}$. After filling the memory with at least $B$ transitions, DeepTOP-MDP updates the parameters $\phi, \theta, \theta'$ in every timestep using a sampled minibatch of size $B$ of transitions $\{s_{t_k}, a_{t_k}, r_{t_k}, s_{t_k+1}\}$, for $1 \le k \le B$. The critic network uses the sampled transitions to calculate the estimated gradient of $\mathcal{L}(\theta)$:

$$\hat{\nabla}_\theta \mathcal{L}(\theta) := \frac{2}{B} \sum_{k=1}^{B} \left( Q^\theta(\lambda_{t_k}, v_{t_k}, a_{t_k}) - r_{t_k} - \gamma \max_{a' \in \mathcal{A}} Q^{\theta'}(\lambda_{t_k+1}, v_{t_k+1}, a') \right) \nabla_\theta Q^\theta(\lambda_{t_k}, v_{t_k}, a_{t_k}). \tag{7}$$

Similarly, the actor network uses the sampled transitions and Equation (3) to calculate the estimated gradient:

$$\hat{\nabla}_\phi K(\mu^\phi) := \frac{1}{B} \sum_{k=1}^{B} \left( Q_{\mu^\phi}^\theta \left( \mu^\phi(v_{t_k}), v_{t_k}, 1 \right) - Q_{\mu^\phi}^\theta \left( \mu^\phi(v_{t_k}), v_{t_k}, 0 \right) \right) \nabla_\phi \mu^\phi(v_{t_k}). \tag{8}$$

---

**Algorithm 1** Deep Threshold Optimal Policy Training for MDPs (DeepTOP-MDP)

---

Randomly select initial actor network parameters $\phi$ and critic network parameters $\theta$.
Set target critic network parameters $\theta' \leftarrow \theta$, and initialize replay memory $\mathcal{M}$.
**for** timestep $t = 1, 2, 3, \ldots$ **do**
    Receive state $s_t = (\lambda_t, v_t)$ from environment $\mathcal{E}$.
    Select action $a_t = \mathbb{1}(\mu^\phi(v_t) > \lambda_t)$ with probability $1 - \epsilon_t$. Otherwise, select action $a_t$ randomly.
    Execute action $a_t$, and observe reward $r_t$ and next state $s_{t+1}$ from $\mathcal{E}$.
    Store transition $\{s_t, a_t, r_t, s_{t+1}\}$ into $\mathcal{M}$.
    Sample a minibatch of $B$ transitions $\{s_{t_k}, a_{t_k}, r_{t_k}, s_{t_k+1}\}$, for $1 \le k \le B$ from $\mathcal{M}$.
    Update critic network parameters $\theta$ using the estimated gradient from Equation (7).
    Update actor network parameters $\phi$ using the estimated gradient from Equation (8).
    Soft update target critic parameters $\theta'$: $\theta' \leftarrow \tau\theta + (1 - \tau)\theta'$.
**end for**

---

Both the critic network and the actor network then take a gradient update step. Finally, we soft update the target critic's parameters $\theta'$ using $\theta' \leftarrow \tau\theta + (1 - \tau)\theta'$, with $\tau < 1$. The complete pseudocode is given in Algorithm 1.

## 4  Whittle Index Policy for RMABs

In this section, we demonstrate how the Whittle index policy [32], a powerful tool for solving the notoriously intractable Restless Multi-Armed Bandit (RMAB) problem, can be represented with a set of threshold functions. We first describe the RMAB control problem, and then define the Whittle index function.

An RMAB problem consists of $N$ arms. The environment of an arm $i$, denoted as $\mathcal{E}_i$, is an MDP with a discrete state space $s_{i,t} \in \mathcal{S}_i$, and a binary action space $a_{i,t} \in \mathcal{A} := \{0, 1\}$, where $a_{i,t} = 1$ means that arm $i$ is activated, and $a_{i,t} = 0$ means that arm $i$ is left passive at time $t$. Given the state-action pair $(s_{i,t}, a_{i,t})$, $\mathcal{E}_i$ generates a random reward $r_{i,t}$ and a random next state $s_{i,t+1}$ following some unknown probability distributions based on $(s_{i,t}, a_{i,t})$. Here we also use $\bar{r}_i(s_i, a_i)$ to denote the unknown expected one-step reward that can be obtained for the state-action pair $(s_i, a_i)$.

A control policy over all arms takes the states $(s_{1,t}, s_{2,t}, \ldots, s_{N,t})$ as input, and activates $V$ out of $N$ arms in every timestep. Solving for the optimal control policy for RMABs was proven to be intractable [21], since the agent must optimize over an input state space exponential in $N$. To circumvent the dimensionality challenge, the Whittle index policy assigns real values to an arm's states using a Whittle index function for each arm $W_i : \mathcal{S}_i \to \mathbb{R}$. Based on the assigned Whittle indices $(W_1(s_{1,t}), W_2(s_{2,t}), \ldots, W_N(s_{N,t}))$, the Whittle index policy activates the $V$ highest-valued arms out of $N$ arms in timestep $t$, and picks the passive action for the remaining arms.

### 4.1  The Whittle Index Function as The Optimal Threshold Function

To define the Whittle index and relate it to threshold functions, let us first consider an alternative control problem of a single arm $i$ as environment $\mathcal{E}_i$ with *activation cost* $\lambda$. In this problem, the agent follows a control policy that determines whether the arm is activated or not based on its current state $s_{i,t}$. If the policy activates the arm, then the agent must pay an activation cost of $\lambda$. Hence, the agent's *net reward* at timestep $t$ is defined as $r_{i,t} - \lambda a_{i,t}$.

We now consider applying threshold policies for this alternative control problem. A threshold policy defines a threshold function $\mu_i : \mathcal{S}_i \to \mathbb{R}$ that maps each state to a real value. It then activates the arm if and only if $\mu_i(s_{i,t}) > \lambda$, i.e., $a_{i,t} = \mathbb{1}(\mu_i(s_i) > \lambda)$. The value of $\mu_i(s_{i,t})$ can therefore be viewed as the largest activation cost that the agent is willing to pay to activate the arm under state $s_{i,t}$. To characterize the performance of a threshold policy with a threshold function $\mu_i(\cdot)$, we let $\rho_{\mu_i,\lambda}(s_i', s_i)$ be the discounted state distribution, which is the average discounted number of visits of state $s_i'$ when the initial state is $s_i$ under the threshold policy and $\lambda$. When the initial state is $s_i$, the expected discounted net reward under the threshold policy is

$$Q_{i,\lambda}(s_i, \mathbb{1}(\mu_i(s_i) > \lambda)) = \sum_{s_i' \in \mathcal{S}_i} \rho_{\mu_i,\lambda}(s_i', s_i)\Big(\bar{r}_i(s_i', \mathbb{1}(\mu_i(s_i') > \lambda)) - \lambda\mathbb{1}(\mu_i(s_i') > \lambda)\Big). \tag{9}$$

The performance of the threshold policy under a given $\lambda$ is defined as $J_{i,\lambda}(\mu_i) := \sum_{s_i \in S_i} Q_{i,\lambda}(s_i, \mathbb{1}(\mu_i(s_i) > \lambda))$. The Whittle index of this arm is defined as the function $\mu_i(\cdot)$ whose corresponding threshold policy maximizes $J_{i,\lambda}(\mu_i)$ for all $\lambda$:

**Definition 1.** *(Whittle Index) If there exists a function $\mu_i : S_i \to \mathbb{R}$ such that choosing $\mathbb{1}(\mu_i(s_i) > \lambda)$ maximizes $J_{i,\lambda}(\mu_i)$ for all $\lambda \in (-\infty, +\infty)$, then we say that $\mu_i(s_i)$ is the Whittle index $W_i(s_i)$* [1].

We note that, for some arms, there does not exist any function $\mu_i(s_i)$ that satisfies the condition in Definition 1. For such arms, the Whittle index does not exist. We say that an arm is *indexable* if it has a well-defined Whittle index function. Definition 1 shows that finding the Whittle index is equivalent to finding the optimal $\mu_i(\cdot)$ that maximizes $J_{i,\lambda}(\mu_i)$ for all $\lambda \in (-\infty, +\infty)$. Parameterizing a threshold function $\mu_i^{\phi_i}(\cdot)$ by parameters $\phi_i$ and letting $M$ be a sufficiently large number such that $\mu_i^{\phi_i}(s_i) \in (-M, +M)$ for all $s_i$ and $\phi_i$, we aim to find the optimal $\phi_i$ for maximizing the objective function

$$K_i(\mu_i^{\phi_i}) := \int_{\lambda=-M}^{\lambda=+M} \sum_{s_i \in S_i} Q_{i,\lambda}(s_i, \mathbb{1}(\mu_i^{\phi_i}(s_i) > \lambda)) d\lambda. \tag{10}$$

# 5  Deep Threshold Optimal Policy for RMABs

To design a DeepTOP variant for RMABs, we first give the gradient of the objective function.

**Theorem 2.** *Given the parameter vector $\phi_i$, let $\bar{\rho}_\lambda(s_i)$ be the discounted state distribution when the initial state is chosen uniformly at random and the activation cost is $\lambda$. If all states $s_i \in S_i$ have distinct values of $\mu_i^{\phi_i}(s_i)$, then,*

$$\nabla_{\phi_i} K_i(\mu_i^{\phi_i}) = |S_i| \sum_{s_i \in S_i} \bar{\rho}_{\mu_i^{\phi_i}(s_i)}(s_i) \Big( Q_{i,\mu_i^{\phi_i}(s_i)}(s_i, 1) - Q_{i,\mu_i^{\phi_i}(s_i)}(s_i, 0) \Big) \nabla_{\phi_i} \mu_i^{\phi_i}(s_i). \tag{11}$$

*Proof.* The proof is similar to that of Theorem 1. For completeness, we provide it in Appendix A.  □

We note that Theorem 2 does not require the arm to be indexable. Whether an arm is indexable or not, using Theorem 2 along with a gradient ascent algorithm will find a locally-optimal $\phi_i$ that maximizes $K_i(\mu_i^{\phi_i})$. When the arm is indexable, the resulting threshold function $\mu_i^{\phi_i}$ is the Whittle index function. Using the gradient result from Equation (11), we present the algorithm DeepTOP-RMAB for finding the optimal parametrized threshold functions $\mu_i^{\phi_i}$ for arms $i = 1, 2, \ldots, N$. The training method is similar to the MDP version, except for two important differences. First, the training of each arm is done independently from others. Second, the value of $\lambda$ is an artificial value that only exists in the alternative problem but not in the original RMAB problem. Similar to DeepTOP-MDP, we maintain three network parameters for each arm $i$: actor $\phi_i$, critic $\theta_i$, and target-critic $\theta_i'$. The critic network parametrizes the action-value function, and is optimized by minimizing the loss function

$$\mathcal{L}_i(\theta_i) := \int_{\lambda=-M}^{\lambda=+M} \mathbb{E}_{s_{i,t}, a_{i,t}, r_{i,t}, s_{i,t+1}} \left[ \Big( Q_{i,\lambda}^{\theta_i}(s_{i,t}, a_{i,t}) - r_{i,t} - \gamma \max_{a' \in \mathcal{A}} Q_{i,\lambda}^{\theta_i'}(s_{i,t+1}, a') \Big)^2 \right] d\lambda, \tag{12}$$

with $(s_{i,t}, a_{i,t}, r_{i,t}, s_{i,t+1})$ sampled under some policy. In each timestep $t$, each arm environment $\mathcal{E}_i$ provides its current state $s_{i,t}$ to the agent. For each arm $i = 1, 2, \ldots, N$, DeepTOP-RMAB calculates the state value $\mu_i^{\phi_i}(s_{i,t})$ with the arm's respective actor network parameters $\phi_i$. Given an exploration parameter $\epsilon_t \in [0, 1)$, DeepTOP-RMAB activates the $V$ arms with the largest $\mu_i^{\phi_i}(s_{i,t})$ with probability $1 - \epsilon_t$, and activates $V$ randomly selected arms with probability $\epsilon_t$. Based on the executed actions, each arm provides a reward $r_{i,t}$ and the next state $s_{i,t+1}$. An arm's transition $\{s_{i,t}, a_{i,t}, r_{i,t}, s_{i,t+1}\}$ is then stored in the arm's memory denoted by $\mathcal{M}_i$. After filling each arm's memory with at least $B$ transitions, DeepTOP-RMAB updates $\phi_i, \theta_i$, and $\theta_i'$ in every timestep. For each arm $i$, DeepTOP-RMAB first samples a minibatch of size $B$ of transitions $\{s_{i,t_k}, a_{i,t_k}, r_{i,t_k}, s_{i,t_k+1}\}$, for $1 \le k \le B$ from the memory $\mathcal{M}_i$. It then randomly samples $B$ values $[\lambda_{i,1}, \lambda_{i,2}, \ldots, \lambda_{i,B}]$ from the range $[-M, +M]$. Using the sampled transitions and $\lambda$ values, it estimates the gradient of $\mathcal{L}_i(\theta_i)$ as

$$\hat{\nabla}_{\theta_i} \mathcal{L}_i(\theta_i) := \frac{2}{B} \sum_{k=1}^{B} \Big( Q_{i,\lambda_k}^{\theta_i}(s_{i,t_k}, a_{i,t_k}) - r_{i,t_k} - \gamma \max_{a' \in \mathcal{A}} Q_{i,\lambda_k}^{\theta_i'}\big(s_{i,t_k+1}, a'\big) \Big) \nabla_{\theta_i} Q_{i,\lambda_k}^{\theta_i}(s_{i,t_k}, a_{i,t_k}). \tag{13}$$

---

[1]To simplify notations, we use a necessary and sufficient condition for the Whittle index as its definition. We refer interested readers to [9] for more thorough discussions on the Whittle index.

Using the sampled transitions and Equation (11), it estimates the gradient of $K_i(\mu_i^{\phi_i})$ as

$$\hat{\nabla}_{\phi_i} K_i(\mu_i^{\phi_i}) := \frac{1}{B} \sum_{k=1}^{B} \left( Q_{i,\mu_i^{\phi_i}(s_{i,t_k})}^{\theta_i} \left( s_{i,t_k}, 1 \right) - Q_{i,\mu_i^{\phi_i}(s_{i,t_k})}^{\theta_i} \left( s_{i,t_k}, 0 \right) \right) \nabla_{\phi_i} \mu_i^{\phi_i}(s_{i,t_k}). \tag{14}$$

A gradient update step is taken after calculating the actor and critic networks' gradients. Finally, DeepTOP-RMAB soft updates the target critic parameters $\theta_i'$ using $\theta_i' \leftarrow \tau\theta_i + (1 - \tau)\theta_i'$, with $\tau < 1$. The complete DeepTOP-RMAB pseudocode is given in Appendix B.

## 6 Simulations

We have implemented and tested both DeepTOP-MDP and DeepTOP-RMAB in a variety of settings. The training procedure of the two DeepTOP algorithms are similar to that of the DDPG [19] algorithm except for the expression of gradients. We implemented the DeepTOP algorithms by modifying an open-source implementation of DDPG [12]. All source code can be found in the repository https://github.com/khalednakhleh/deeptop.

### 6.1 Simulations for MDPs

We evaluate three MDPs, namely, the electric vehicle charging problem, the inventory management problem, and the make-to-stock problem.

**EV charging problem.** This problem is based on Yu, Xu, and Tong [34]. It considers a charging station serving EVs. When an EV arrives at the station, it specifies the amount of charges it needs and a deadline upon which it will leave the station. The electricity price changes over time and we model it by an Ornstein-Uhlenbeck process [30]. In each timestep, the station decides whether to charge the EV or not. If it decides to charge the EV, then it provides one unit charge to the EV. The station then obtains a unit reward and pays the current electricity price. If the station fails to fully charge the EV by the deadline of the EV, then the station suffers from a penalty that is a convex function of the remaining needed charge. A new EV arrives at the station when the previous EV leaves. We model this problem by letting the scalar state be the current electricity price and the vector state be the remaining needed charge and time-to-deadline of the current EV. A threshold policy is one that calculates a threshold based on the EV's remaining needed charge and time-to-deadline, and then decides to charge the EV if and only if the current electricity price is below the threshold.

**Inventory management problem.** We construct an inventory management problem by jointly incorporating a variety of practical challenges, including seasonal fluctuations in demands and lead times in orders, in the literature [28, 15, 10, 27]. We consider a warehouse holding goods. In each timestep, there is a random amount of demand whose mean depends on the time of the year. The warehouse can fulfill the demand as long as it has sufficient inventory, and it makes a profit for each unit of sold goods. At the end of the timestep, the warehouse incurs a unit holding cost for each unit of unsold goods. The warehouse manager needs to decide whether to order more goods. When it places an order for goods, there is a lead time of one time step, that is, the goods ordered at timestep $t$ are only available for sale at timestep $t + 1$. We model this problem by letting the scalar state be the current inventory and the vector state be the time of the year. A threshold policy calculates a threshold based on the time of the year and decides to place an order for goods if the current inventory is below the threshold.

**Make-to-stock production problem.** This problem is considered in [26]. It studies a system that produces $m$ items with $W$ demand classes and buffer size $S$. Accepting a class $v$ order leads to a reward $R_v$, as long as there is still room in the buffer for the order. The classes of demands are ordered such that $R_1 > R_2 > \ldots$. In this problem, the scalar state is the number of accepted but unfinished orders and the vector state is the class of the next arriving order. More details about the three MDPs can be found in Appendix C.

**Evaluated policies.** We compare DeepTOP-MDP against DDPG [19] and TD3 [8], two state-of-the-art off-policy and model free deep RL algorithms. We use open-source implementations of these two algorithms for [12, 7]. We use the same hyper-parameters, including the neural network architecture, learning rates, etc., for all three algorithms. We also evaluate the Structure-Aware Learning for Multiple Thresholds algorithm (SALMUT) [26], a reinforcement learning algorithm

that finds the optimal threshold policy. SALMUT requires the vector states to be pre-sorted by their threshold values. Hence, SALMUT can only be applied to the make-to-stock production problem. Details about the training parameters can be found in Appendix D. For the EV charging problem, Yu, Xu, and Tong [34] has found the optimal threshold policy. We call the optimal threshold policy the *Deadline Index* policy and compare DeepTOP-MDP against it.

**Simulations results.** Simulation results of the three MDPs are shown in Figure 1. The results are the average of 20 independent runs. Before starting a run, we fill an agent's memory with 1000 transitions by randomly selecting actions. We plot the average reward obtained from the previous 100 timesteps, and average them over 20 runs. In addition, we provide the standard deviation bounds from the average reward.

It can be observed that DeepTOP significantly outperforms DDPG, TD3, and SALMUT. Although the training procedure of DeepTOP is similar to that of DDPG, DeepTOP is able to achieve much faster learning by leveraging the monotone property. Without leveraging the monotone property, DDPG and TD3 need to learn the optimal policy for each scalar state independently, and therefore have much worse performance. DeepTOP performs better than SALMUT because DeepTOP directly employs the threshold policy gradient. SALMUT in contrast approximates threshold policies through randomized policies since it can only handle continuous and differentiable functions. We believe this might be the reason why DeepTOP outperforms SALMUT. We also note that DeepTOP performs virtually the same as the Deadline Index policy for the EV charging problem in about 2000 timesteps, suggesting that DeepTOP indeed finds the optimal threshold policy quickly. We also evaluate DeepTOP for different neural network architectures in Appendix E, and show that DeepTOP performs the best in all settings.

## 6.2 Simulations for RMABs

We evaluate two RMABs, namely, the one-dimensional bandits from [17] and the recovering bandits from [20].

**One-dimensional bandits.** We consider an extension of the RMAB problem evaluated in Killian et al. [17]. Killian et al. [17] considers the case when each arm is a two-state Markov process. We extend it so that each arm is a Markov process with 100 states, numbered as $0, 1, \ldots, 99$, as shown in Figure 2 where state 99 is the optimal state.

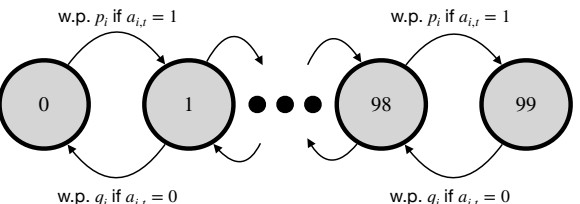

Figure 2: Arm $i$ as a Markov process with 100 states and transition probabilities $p_i$ and $q_i$.

The reward of an arm depends on the distance between its current state and state 99. Suppose the current state of arm $i$ is $s_{i,t}$, then it generates a reward $r_{i,t} = 1 - (\frac{s_{i,t} - 99}{99})^2$. If the arm is activated, then it changes to state $s_{i,t+1} = \min\{s_{i,t} + 1, 99\}$ with probability $p_i$. If the arm is not activated, then it changes to state $s_{i,t+1} = \max\{s_{i,t} - 1, 0\}$ with probability $q_i$. In the simulations, we pick the probabilities $p_i$ to be evenly spaced depending on the number of arms $N$ from the interval $[0.2, 0.8]$. We set the

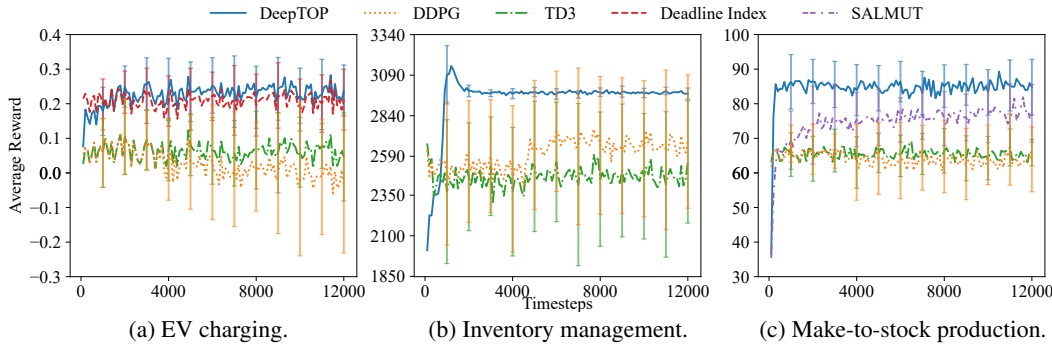

(a) EV charging.  (b) Inventory management.  (c) Make-to-stock production.

Figure 1: Average reward results for the MDP problems.

probabilities $q_i = p_i$. We consider that there are $N$ arms and that the agent needs to activate $V$ arms in each timestep. We evaluate three settings of $(N, V) = (10, 3), (20, 5),$ and $(30, 6)$.

**Recovering bandits.** First introduced in [25], we consider the case that studies the varying behavior of consumers over time. A consumer's interest in a particular product falls if the consumer clicks on its advertisement link. However their interest in the product would recover with time. The recovering bandit is modelled as an RMAB with each arm being the advertisement link. The reward of playing an arm is given by a function $f(\min(z, z_{max}))$, with $z$ being the time since the arm was last played.

In our experiments, we consider arms with different reward functions, with the arm's state being the value $\min\{z, z_{max}\}$ and $z_{max} = 100$. We also evaluate recovering bandits on three settings of $(N, V) = (10, 3), (20, 5),$ and $(30, 6)$. More details can be found in Appendix F.

**Evaluated policies.** We compare DeepTOP-RMAB against three recent studies that aim to learn index policies for RMABs, namely, Lagrange policy Q learning (LPQL) [17], Whittle index based Q learning (WIBQL) [1], and neural Whittle index network (NeurWIN) [20]. LPQL consists of three steps: First, it learns a Q function for each arm independently. Second, it uses the Q functions of all arms to determine a common Lagrangian. Third, it uses the Lagrangian to calculate the index of each arm. WIBQL is a two-timescale algorithm that learns the Whittle indices of indexable arms by updating Q values on the fast timescale, and index values on the slower timescale. NeurWIN is an off-line training algorithm based on REINFORCE that requires a simulator to learn the Whittle index. Both LPQL and WIBQL are tabular learning methods which may perform poorly compared to deep RL algorithms when the size of the state space is large. Hence, we also design deep RL equivalent algorithms that approximate their Q functions using neural networks. We refer to the Deep RL extensions as neural LPQL and neural WIBQL. In all experiments, neural LPQL, neural WIBQL, and NeurWIN use the same hyper-parameters as DeepTOP-RMAB. For the one-dimensional bandits, it can be shown that the Whittle index is in the range of $[-1, 1]$, and hence we set $M = 1$. For the recovering bandits, we set $M = 10$.

**Simulation results.** Simulation results are shown in Figures 3 and 4. It can be observed that DeepTOP achieves the optimal average rewards in all cases. The reason that neural LPQL performs worse than DeepTOP may lie in its reliance on a common Lagrangian. Since the common Lagrangian is calculated based on the Q functions of all arms, an inaccuracy in one arm's Q function can result in an inaccurate Lagrangian, which, in turn, leads to inaccuracy in the index values of all arms. Prior work [17] has already shown that WIBQL performs worse than LPQL. Hence, it is not surprising that neural WIBQL performs worse than both neural LPQL and DeepTOP. NeurWIN performs worse than DeepTOP because it is based on REINFORCE and therefore can only apply updates at the end of each minibatch of episodes. We also evaluate DeepTOP for different neural network architectures and the results are shown in Appendix G for the one-dimensional bandits and Appendix H for the recovering bandits.

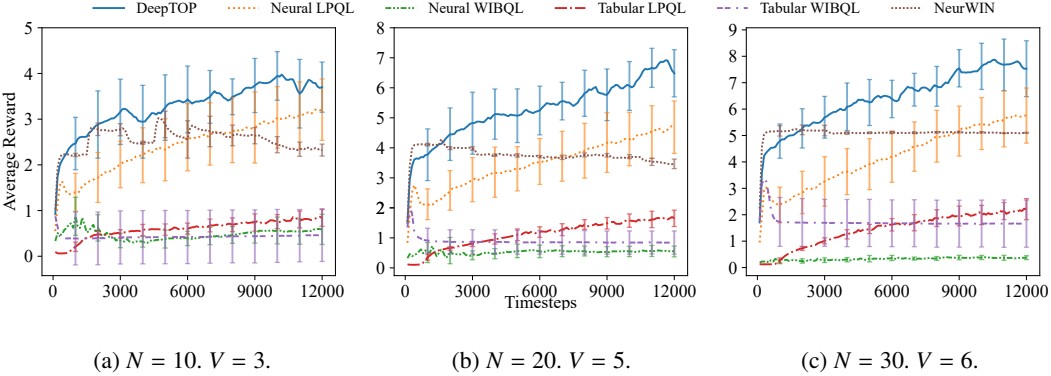

(a) $N = 10. V = 3.$      (b) $N = 20. V = 5.$      (c) $N = 30. V = 6.$

Figure 3: Average reward results for the one-dimensional bandits.

## 7 Related Work

Threshold policies have been analysed for many decision-making problems formed as MDPs. [11] examined the residential energy storage under price fluctuations problem, and proved the existence

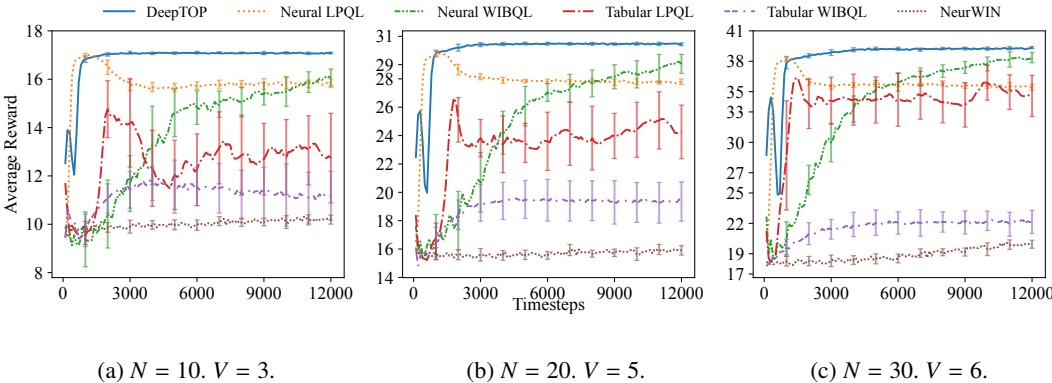

(a) $N = 10$. $V = 3$.    (b) $N = 20$. $V = 5$.    (c) $N = 30$. $V = 6$.

Figure 4: Average reward results for the recovering bandits.

of optimal threshold policies for minimizing the cost. [5] proved that MDPs with a convex and piecewise linear cost functions admit an optimal threshold policy. [24] shows the existence of an optimal threshold policy for energy arbitrage given degrading battery capacity, with [2] using the REINFORCE algorithm [33] to learn a trading policy with price thresholds for intraday electricity markets. [14] considered mean field games in a multi-agent MDP setting, and characterized individual agent strategy with a threshold policy when the mean game admits a threshold policy.

More recently, [31] studies finding a job assigning threshold policy for data centers with heterogeneous servers and job classes, and gave conditions for the existence of optimal threshold policies. [35] proposed a distributed threshold-based control policy for graph traversal by assigning a state threshold that determines if the agent stays in or leaves a state. For minimizing the age of information in energy-harvesting sensors, [4] used the finite-difference policy gradient [23] to learn a possibly sub-optimal threshold policy in the average cost setting. [13] proposed an RL-based threshold policy for semi-MDPs in controlling micro-climate for buildings with simulations proving efficacy on a single-zone building. [29] used the Deep Q-network RL algorithm for selecting alert thresholds in anti-fraud systems with simulations showing performance improvements over static threshold policies. [26] described the SALMUT RL algorithm for exploiting the ordered multi-threshold structure of the optimal policy with SALMUT implementations in [16] for computing node's overload protection. In contrast to these works, DeepTOP-MDP is applicable to any MDP that admits threshold policies.

In learning the Whittle index policy for RMABs, [6] proposed a Q-learning heuristic called the Q Whittle Index Controller (QWIC) which may not find the Whittle indices even when the training converges. [20] describes a Deep RL algorithm called NeurWIN for learning the Whittle index of a restless arm independently of other arms. However, NeurWIN requires a simulator to train the neural networks. Some recent studies, such as [1, 3, 17], proposed various online learning algorithms that can find Whittle index when the algorithms converge. These algorithms rely on some indirect property of the Whittle index which explains why they converge slower than DeepTOP.

## 8 Conclusion and Future Work

In this paper, we presented DeepTOP: a Deep RL actor-critic algorithm that learns the optimal threshold function for MDPs that admit a threshold policy and for RMAB problems. We first developed the threshold policy gradient theorem, where we proved that a threshold function has a simple to compute gradient. Based on the gradient expressions, we design the DeepTOP-MDP and DeepTOP-RMAB algorithm variants and compare them against state-of-the-art learning algorithms. In both the MDP and RMAB settings, experiment results showed that DeepTOP exceeds the performance of baselines in all considered problems. A promising future direction is to extend DeepTOP to threshold policies with multiple actions. For example, the Federal Reserve needs to decide not only whether to raise interest rate, but also the amount of rate hike.

## Acknowledgments and Disclosure of Funding

This material is based upon work supported in part by NSF under Award Number ECCS-2127721, in part by the U.S. Army Research Laboratory and the U.S. Army Research Office under Grant Number W911NF-22-1-0151, and in part by Office of Naval Research under Contract N00014-21-1-2385. Portions of this research were conducted with the advanced computing resources provided by Texas A&M High Performance Research Computing.

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
