# OpenReview forum: "DeepTOP: Deep Threshold-Optimal Policy for MDPs and RMABs"
_NeurIPS.cc/2022/Conference — NeurIPS 2022 Accept_

### Official Review · Reviewer_bZdV · 2022-07-03

**Rating:** 6
**Confidence:** 3
**Soundness:** 4 excellent
**Presentation:** 4 excellent
**Contribution:** 2 fair

**Summary:**

The paper considers threshold policies problem. The authors show that the gradient for these problems has a simple expression. The authors also propose a rephrasing of Whittle index policies for restless multi-armed bandits in the form of threshold policy and match their algorithm to this scenario. The authors support their results with simulations.

**Questions:**

The paper is pretty straight-forward and clear. I have no questions or suggestions.

**Limitations:**

The paper has no potential negative societal impact.

**Strengths And Weaknesses:**

Originality
To the best of my knowledge the results are novel.

Quality
The theoretical results do not seem very surprising, but I did find them interesting and useful.

Clarity
The paper is written very clearly.
The restrictions on V seems to be quite drastic (discrete set at line 66, distinct threshold values in Theorem 1) so I think a short explanation is in order (is this just a technical restriction or a real pain?, why the restriction exists?)

Significance
Threshold policies seem to match a rather small range of problems. The main limitations are two actions policies and the policy structure.  Subsequently, the significance of the paper is highly limited just by tackling this small range. In its niche, I think the paper gives a very useful insight, even if its not very sophisticated. In addition, with some thought the core idea might extend to more general scenarios. For example - other cases where the gradient can be calculated easily or other problems that can be solved to threshold policies.

---

> ### Author Response · Authors · 2022-08-01
> **Response to reviewer bZdV comments**
>
> We thank the reviewer for their detailed comments. If our responses are satisfactory, we kindly ask the reviewer to update the score.
>
>
> ### *The restrictions on V seems to be quite drastic (discrete set at line 66, distinct threshold values in Theorem 1) so I think a short explanation is in order (is this just a technical restriction or a real pain?, why the restriction exists?)*
>
> The assumption is needed for the first step of the proof, where we decompose the region $[-M, M]$ into several distinct intervals.
>
> In practice, this assumption is not a serious restriction. For any randomly initialized neural networks, it is near impossible to have the same outputs for two different inputs during any update sequences.
>
>
> ### *The main limitations are two actions policies and the policy structure. Subsequently, the significance of the paper is highly limited just by tackling this small range. In its niche, I think the paper gives a very useful insight, even if its not very sophisticated. In addition, with some thought the core idea might extend to more general scenarios. For example - other cases where the gradient can be calculated easily or other problems that can be solved to threshold policies.*
>
> We thank the reviewer for the insight. We are indeed working on extending this work for multi-action threshold policies. In this case, the neural network outputs  multiple threshold values $\lambda_k, \lambda_{k+1},...$. The policy deterministically picks action $a_k$ if the scalar state lies in the interval between $\lambda_k$ and $\lambda_{k+1}$.
>
> We would also like to emphasize that there are many problems, especially in the queueing and networking community, where it is natural to consider threshold policies, and, in many cases, the optimal policies are indeed threshold ones. For example:
>
> [36] H. Tang, J. Wang, L. Song and J. Song, "Minimizing Age of Information With Power Constraints: Multi-User Opportunistic Scheduling in Multi-State Time-Varying Channels," in IEEE Journal on Selected Areas in Communications, vol. 38, no. 5, pp. 854-868, May 2020, doi: 10.1109/JSAC.2020.2980911.
>
> [37] B. Zhou and W. Saad, "Minimum Age of Information in the Internet of Things With Non-Uniform Status Packet Sizes," in IEEE Transactions on Wireless Communications, vol. 19, no. 3, pp. 1933-1947, March 2020, doi: 10.1109/TWC.2019.2959777.
>
> [38] Eitan Altman, Rachid El-Azouzi, Daniel Sadoc Menasche, and Yuedong Xu. 2019. Forever Young: Aging Control For Hybrid Networks. In Proceedings of the Twentieth ACM International Symposium on Mobile Ad Hoc Networking and Computing (Mobihoc '19). Association for Computing Machinery, New York, NY, USA, 91–100. https://doi.org/10.1145/3323679.3326507.
>
> [39] Guidan Yao, Ahmed M. Bedewy, and Ness B. Shroff. 2021. Battle between Rate and Error in Minimizing Age of Information. In Proceedings of the Twenty-second International Symposium on Theory, Algorithmic Foundations, and Protocol Design for Mobile Networks and Mobile Computing (MobiHoc '21). Association for Computing Machinery, New York, NY, USA, 121–130. https://doi.org/10.1145/3466772.3467041.
>
> [40] T. Z. Ornee and Y. Sun, "Sampling and Remote Estimation for the Ornstein-Uhlenbeck Process Through Queues: Age of Information and Beyond," in IEEE/ACM Transactions on Networking, vol. 29, no. 5, pp. 1962-1975, Oct. 2021, doi: 10.1109/TNET.2021.3078137.

---

### Official Review · Reviewer_dpbM · 2022-07-07

**Rating:** 5
**Confidence:** 4
**Soundness:** 3 good
**Presentation:** 3 good
**Contribution:** 1 poor

**Summary:**

The paper presents an algorithm to compute an optimal threshold policy in MDPs and RMABs with state information composed of a scalar state and a vector state. The authors propose to learn a mapping from the vector state to a scalar number to compare with the scalar state. This function is used to construct the threshold policy where the action (0 or 1) only depends on the comparison between the scalar state and the produced scalar value. In order to learn the mapping used for threshold policy, the authors use an actor-critic algorithm, where the scalar mapping and the associated threshold policy are used as the actor function, and a neural network is used as the action-value function (Q-function) as the critic function. The losses of the actor and critic functions are defined as the standard actor-critic work using the expected performance and the Bellman error. In this paper, the authors compute the derivative of the expected performance and identify a simple expression of the actor's derivative. This is used to perform actor-critic gradient updates more efficiently.

In the RMAB domain, the same algorithm and derivative simplification can be applied to the RMAB domain. RMABs is a special case of multiple MDPs with scalar and vector states. Specifically, the objective is defined as the integral of all activation cost $lambda$, assuming Whittle index exists and thus there exists a threshold policy that is optimal for all activation cost. This makes finding the optimal threshold policy equivalent to finding the Whittle index (if exist).

The proposed method is evaluated on three domains and compared with other RL-based baselines, including general RL algorithms (DDPG, TD3, SALMUT) in the MDP setting, and Q-learning based (LPQL, WIBQL) and Whittle index based (NeurWIN) in the RMABs setting. My interpretation of why the proposed algorithm can outperform the general RL-based algorithms is that the proposed method simplifies the search space to threshold policy, while in contrast the general RL algorithms may use more complex models (e.g., neural networks) to represent the actor function. This advantage makes the proposed algorithm find the optimal policy more efficient but also restricted more to threshold policy. It may not work when threshold policy is not optimal. Specifically, in the context of MDPs considered in this paper, it is possible that threshold policy is not optimal. In those cases, general RL algorithms may still be needed.

in the RMABs context, the proposed algorithm outperforms other Q-learning based algorithm without using actor-critic algorithm. It is known that actor-critic can improve the performance of the RL challenges. I believe this is the main advantage of the proposed algorithm compared to other baselines.

**Questions:**

## Comments
- I think there is a missing integral over $\lambda'$ in the definition of Q function in Equation (1).

## Questions
- I understand that finding the optimal policy in MDPs and RMABs are both challenging due to the PSPACE hardness. But finding the Whittle index in RMABs may be polynomial time solvable when there are only finitely many states [34, 35], where given indexability, [34] uses the definition of Whittle index and the Bellman equation to form a LP to solve in polynomial time, and [35] leverages the threshold policy to construct a faster algorithm for a specific type of RMABs problems. Is it possible to directly compute the Whittle index using similar LP method without using RL or neural networks? If not, what is the major difficulty of computing the Whittle index directly in your case?
- Why did you choose to use a specific quadratic form of the reward function in the RMAB simulation in Section 6.3? Does the reward function structure affect the convergence of the actor-critic gradient descent update?


References:
[34] Qian, Yundi, Chao Zhang, Bhaskar Krishnamachari, and Milind Tambe. "Restless poachers: Handling exploration-exploitation tradeoffs in security domains." In Proceedings of the 2016 International Conference on Autonomous Agents & Multiagent Systems, pp. 123-131. 2016.
[35] Mate, Aditya, Jackson Killian, Haifeng Xu, Andrew Perrault, and Milind Tambe. "Collapsing Bandits and Their Application to Public Health Intervention." Advances in Neural Information Processing Systems 33 (2020): 15639-15650.



**Limitations:**

## Limitations
- [Stated by the authors] The algorithm is only applicable to MDPs that admit a threshold policy.
- The proposed algorithm only works with threshold policy with a single scalar value.

## Negative societal impact
N/A

**Strengths And Weaknesses:**

## Strengths
- The paper is well-presented and easy to follow. I appreciate the clarity of the presentation and idea.
- The simplified expression of the actor derivative (expected reward derivative) is new.
- Thorough evaluations and experiments

--------
## Weaknesses
- The novelty is incremental. The main contribution is based on the use of threshold policy and simplification of the policy gradient.
- The MDPs and RMABs domains are similar with no major differences.
- The threshold policy considered in the paper can only handle one single scalar, which limits the applicability of the threshold policy.
- [Minor]The proposed policy only works when threshold policy is good enough. Otherwise, a more expressive policy parameterization is still needed in order to achieve better performance.

---

> ### Author Response · Authors · 2022-08-01
> **Response to reviewer dpbM comments - part 1**
>
> We thank the reviewer for their detailed comments. If our responses are satisfactory, we kindly ask the reviewer to update their score.
>
> ### *In the RMABs context, the proposed algorithm outperforms other Q-learning based algorithm without using actor-critic algorithm. It is known that actor-critic can improve the performance of the RL challenges. I believe this is the main advantage of the proposed algorithm compared to other baselines.*
>
> We would like to emphasize that, for RMAB, we have indeed evaluated other Q-learning algorithms with actor-critic implementations. They are called "Neural LPQL" and "Neural WIBQL" in Fig. 3. Neural LPQL, Neural WIBQL, and our DeepTOP have the same implementation of the critic network. They only differ in the update rule for the actor. Fig. 3 shows that DeepTOP significantly outperforms Neural LPQL and Neural WIBQL. This shows that the superiority of DeepTOP is not only due to actor-critic networks, but also due to its better update rule for the actor. Below, we explain in details why Neural LPQL and Neural WIBQL perform worse than DeepTOP.
>
> Neural LPQL operates as follows: Given the critic networks of all bandits, LPQL use all of them together to find a single Lagrange multiplier. LPQL then updates the actor network to find the optimal index of each individual bandit under this Lagrange multiplier. The problem of this approach is that the calculation of index of bandits is not independent from each other. If the critic network of a single bandit is far off, then it will cause LPQL to obtain a wrong Lagrange multiplier and results in the wrong indexes of all bandits. In contrast, DeepTOP train each bandit completely independently from each other. This ensures that the inaccuracy of one bandit will not propagate to other bandits.
>
> Neural WIBQL operates as follows: Given the critic network Q of one bandit, WIBQL aims to update the actor so that $Q(k,0)-Q(k,1) = 0$ (Eq. (11) in [1]) for each state k independently. This is less direct and efficient than our DeepTOP.
>
> Another important limitation of WIBQL is that it requires each bandit to be indexable. In contrast, our Theorem 2 does not require the bandit to be indexable. Even if a bandit is not indexable, DeepTOP is guaranteed to find a locally optimal threshold policy.
>
>
> ### *The MDPs and RMABs domains are similar with no major differences.*
>
> We agree that extending DeepTOP for RMABs is not difficult. However, we would like to note that RMAB is a very important field of study. The fact that our DeepTOP can be easily applied to RMABs should be considered as a major strength.
>
> In addition, we would like to report that we have conducted experiments for another RMAB problem. In particular, we have evaluated the recovering bandit problem described in [19]. The results are shown in figure 7 in the uploaded rebuttal version. It can be seen that DeepTOP achieves the best performance compared to the baselines.
>
> ### *The threshold policy considered in the paper can only handle one single scalar, which limits the applicability of the threshold policy. The proposed algorithm only works with threshold policy with a single scalar value.*
>
> We thank the reviewer for the comment. We believe it is possible to extend this work to more sophisticated policies. For example, one extension we are considering is multi-action threshold policies. In this case, the neural network outputs multiple threshold values $\lambda_k, \lambda_{k+1},...$. The policy deterministically picks action $a_k$ if the scalar state lies in the interval between $\lambda_k$ and $\lambda_{k+1}$.
>
> Similarly, we can also consider systems where the state consists of a vector of scalars. The threshold policy would output a vector of thresholds, and the action taken depends on which scalar state is above its corresponding threshold.

---

> > ### Author Response · Authors · 2022-08-01
> > **Response to reviewer dpbM comments - part 2**
> >
> > ### *[Stated by the authors] The algorithm is only applicable to MDPs that admit a threshold policy.* and
> > ### *[Minor]The proposed policy only works when threshold policy is good enough. Otherwise, a more expressive policy parameterization is still needed in order to achieve better performance.*
> >
> > Yes, our contribution is limited to threshold policies. However, we would like to note that there are many problems, especially in the queueing and networking community, where it is natural to consider threshold policies, and, in many cases, the optimal policies are indeed threshold ones. For example:
> >
> > [36] H. Tang, J. Wang, L. Song and J. Song, "Minimizing Age of Information With Power Constraints: Multi-User Opportunistic Scheduling in Multi-State Time-Varying Channels," in IEEE Journal on Selected Areas in Communications, vol. 38, no. 5, pp. 854-868, May 2020, doi: 10.1109/JSAC.2020.2980911.
> >
> > [37] B. Zhou and W. Saad, "Minimum Age of Information in the Internet of Things With Non-Uniform Status Packet Sizes," in IEEE Transactions on Wireless Communications, vol. 19, no. 3, pp. 1933-1947, March 2020, doi: 10.1109/TWC.2019.2959777.
> >
> > [38] Eitan Altman, Rachid El-Azouzi, Daniel Sadoc Menasche, and Yuedong Xu. 2019. Forever Young: Aging Control For Hybrid Networks. In Proceedings of the Twentieth ACM International Symposium on Mobile Ad Hoc Networking and Computing (Mobihoc '19). Association for Computing Machinery, New York, NY, USA, 91–100. https://doi.org/10.1145/3323679.3326507.
> >
> > [39] Guidan Yao, Ahmed M. Bedewy, and Ness B. Shroff. 2021. Battle between Rate and Error in Minimizing Age of Information. In Proceedings of the Twenty-second International Symposium on Theory, Algorithmic Foundations, and Protocol Design for Mobile Networks and Mobile Computing (MobiHoc '21). Association for Computing Machinery, New York, NY, USA, 121–130. https://doi.org/10.1145/3466772.3467041.
> >
> >
> > [40] T. Z. Ornee and Y. Sun, "Sampling and Remote Estimation for the Ornstein-Uhlenbeck Process Through Queues: Age of Information and Beyond," in IEEE/ACM Transactions on Networking, vol. 29, no. 5, pp. 1962-1975, Oct. 2021, doi: 10.1109/TNET.2021.3078137.
> >
> >
> > ### *I think there is a missing integral over $\lambda'$ in the definition of Q function in Equation (1).*
> >
> > We thank the reviewer for the correction. We have updated the expression in the uploaded rebuttal version to be $ Q_{\mu}\Big(\lambda, v, 1(\mu(v) > \lambda)\Big) = \sum_{v'\in \mathcal{V}}  \int_{\lambda' = -M}^{\lambda' = +M}\rho_{\mu}(\lambda',v', \lambda, v)\bar{r}\big(\lambda',v', 1(\mu(v') > \lambda')\big).$
> >
> > ### *I understand that finding the optimal policy in MDPs and RMABs are both challenging due to the PSPACE hardness. But finding the Whittle index in RMABs may be polynomial time solvable when there are only finitely many states [34, 35], where given indexability, [34] uses the definition of Whittle index and the Bellman equation to form a LP to solve in polynomial time, and [35] leverages the threshold policy to construct a faster algorithm for a specific type of RMABs problems. Is it possible to directly compute the Whittle index using similar LP method without using RL or neural networks? If not, what is the major difficulty of computing the Whittle index directly in your case?*
> >
> > Yes, it is possible to directly calculate the Whittle index using algorithms in [34,35]. However, these algorithms require the knowledge of the transition kernel or a good estimate of it. In contrast, our algorithms learn the optimal action without the transition kernel in a model-free fashion.
> >
> > Moreover, [34,35] require the bandits to be indexable. In contrast, our Theorem 2 makes no assumptions on indexability. Even when bandits are not indexable, our DeepTOP is guaranteed to find a locally optimal threshold policy.
> >
> >
> > ### *Why did you choose to use a specific quadratic form of the reward function in the RMAB simulation in Section 6.3? Does the reward function structure affect the convergence of the actor-critic gradient descent update?*
> >
> > There is no particular reason for using the quadratic form. To show that the reward function structure does not impact the performance much, we trained DeepTOP and the baselines on a linear reward function $1 - \frac{(99 - s_{i,t})}{99}$ and a cubic reward function $1 - (\frac{(99-s_{i,t})}{99})^3$.
> >
> > We provide the results in the updated rebuttal version in figures 4 (for linear reward function) and figure 5 (for cubic reward function).
> > In both cases, DeepTOP still outperforms the baselines, and gives a superior performance for the three reward functions (quadratic, linear, cubic).

---

> > > ### Comment · Reviewer_dpbM · 2022-08-09
> > > **Thank you for the clarifications**
> > >
> > > Thank you for providing the detailed response. I agree that the benefit of DeepTOP is not just due to the benefit of using actor-critic model but also the update rule (a faster way to compute policy gradient as the main contribution of the paper). One small confusion I have in mind ( sorry for my late response so I understand there may not be enough time to respond) is that the x-axis in Figure 3 is the number of timesteps run by each algorithm, which if the authors also use a actor-critic method (without using the faster policy gradient), it should perform almost identical to the DeepTOP algorithm. I think the benefit of DeepTOP is on the computation but not on the performance (although they are correlated). My understanding is that the faster policy gradient update rule (the contribution of this paper) makes actor-critic style algorithm possible, and thus leads to better performance.
> > >
> > > If possible, could you also briefly explain why there was no actor-critic style algorithm in the literature? Is the computation the major bottleneck of the actor-critic style algorithm? Thank you!

---

> > > > ### Author Response · Authors · 2022-08-09
> > > > **Response to rebuttal discussion**
> > > >
> > > > Thank you for the comments. For the first question:
> > > >
> > > > We agree that the following statements are true:
> > > > 1. If all three actor-critic algorithms use the same rule for updating the actor network, then they become the same and would have exactly the same performance.
> > > > 2. If we run the algorithms for a much longer time, then Neural LPQL and Neural WIBQL would eventually offer the same performance as our DeepTOP.
> > > >
> > > > Hence, we also agree that the benefit of DeepTOP over the other two algorithms is that its gradient update for the actor network is more efficient. This allows it to converge faster by achieving a near-optimal performance with a smaller number of time steps, or, equivalently, achieving a better performance when the number of time steps is small.
> > > >
> > > > For the second question:
> > > >
> > > > The study of using ML for RMAB is still in its infancy. All baseline policies we found were published in the last two years. This is why there are no actor-critic algorithms for this problem in the literature.

---

### Official Review · Reviewer_1kVF · 2022-07-11

**Rating:** 6
**Confidence:** 3
**Soundness:** 3 good
**Presentation:** 2 fair
**Contribution:** 2 fair

**Summary:**

The paper consider a subset of dynamic problems, in which the optimal policy is a threshold-policy. The authors use this attribute to formulate tailored off-policy actor-critic algorithms, for both MDPs and RMABs which are gradient-based, so can utilize neural networks. They empirically compare their method to SOTA methods in three MDP domains and three RMAB parametrizations, the results show that their method, DeepTOP, performs better than the compared methods in all the experiments.

**Questions:**

1. Would you be able to run experiments in more complex domains?
2. Is it possible to give a similar analysis for any policy which is a fixed, deterministic function of some scalar $\lambda_t$ and an output of a neural-network? might be a future direction.

**Limitations:**

The authors addressed the limitations of their work.

**Strengths And Weaknesses:**

Overall, I think it is a good paper, which contributes to the community. But I do have concerns regarding the empirical experiments, I think that the environments are rather toy problems, and since DeepTop incorporate neural-networks, its main advantage over tailored analytical methods is in complex environments.

# Strengths:
The performance of DeepTOP compared to the other methods is impressive. I think that while being limited, threshold policies are indeed interesting. The theorems are important contributions as well.

# Weaknesses:
1. The main contribution of the paper is an empirical method, and the experiments conducted in simple domains. I think that domains that are more challenging should be considered.
2. While being important, the theorems are minor, hence they do not compensate lack of experiments

# Minor Comments:
The last part of section 6 seems like it addressed to the reviewers (lines 220-221). Rephrase.

---

> ### Author Response · Authors · 2022-08-01
> **Response to reviewer 1kVF comments**
>
> We thank the reviewer for their detailed comments. If our responses are satisfactory, we kindly ask the reviewer to update the score.
>
> ### *1. But I do have concerns regarding the empirical experiments, I think that the environments are rather toy problems, and since DeepTop incorporate neural-networks, its main advantage over tailored analytical methods is in complex environments.*
>
> We would like to report that we have conducted experiments for an RMAB with a more complicated setting. In particular, we have evaluated the recovering bandit problem described in [19]. The results are shown in figure 7 in the uploaded rebuttal version. It can be seen that DeepTOP achieves the best performance compared to the baselines.
>
> When designing the environments, our goal was to use the same, or very similar, environments to those employed in the baseline policies. This is why we chose the EV problem from [32], the make-to-stock problem from [24] for MDP experiments, and why we extend the two-state process from [16] to the 100-state process for RMAB experiments.
>
> ### *2. The last part of section 6 seems like it addressed to the reviewers (lines 220-221). Rephrase.*
>
> We have rephrased the sentence in the updated rebuttal version. We will link the code repository after the rebuttal period.
>
> ### *3. The main contribution of the paper is an empirical method, and the experiments conducted in simple domains. I think that domains that are more challenging should be considered* and *4. Would you be able to run experiments in more complex domains?*
>
> Since Theorem 2 holds for all RMABs, DeepTOP is naturally applicable to more complicated settings. The new results on the recovering bandits' setting in figure 7 show that DeepTOP outperforms the baselines.
> We chose the original RMAB simulation setting because it is a more complicated version of the setting used in [16], and we wanted to use a setting similar to [16]. Based on the request of another reviewer, we also evaluated the case when the reward function is $1 - (\frac{99 - s_{i,t}}{99})$ and $1 - (\frac{99-s_{i,t}}{99})^3$. Results in the updated paper (figures 4 and 5) show that our DeepTOP algorithm is still better.
>
> ### *5. Is it possible to give a similar analysis for any policy which is a fixed, deterministic function of some scalar $\lambda_t$ and an output of a neural-network? might be a future direction.*
>
> We thank the reviewer for the insight. This is indeed a promising future direction.
>
> We are actually working on a special case of this direction. We are considering expanding the threshold policy gradient theorem to the case with multiple actions. In this case, the neural network outputs multiple threshold values $\lambda_k, \lambda_{k+1}, ... $. The policy deterministically picks action $a_k$ if the scalar state lies in the interval between $\lambda_k$ and $\lambda_{k+1}$.

---

> > ### Comment · Reviewer_1kVF · 2022-08-09
> > **Thanks for the clarifications**
> >
> > I thank the authors for their detailed response and the revised version. As I said in my review, I think it is a good paper, which has several contributions (both theoretical and empirical). I believe that each contribution as itself is minor, but all of them together are good enough.
> > I still think that more complex empirical experiments could improve the overall contribution (one of the advantages of DeepTOP is the ability to tackle more complex settings than the baselines, maybe use different baselines for the complex experiment?).

---

### Official Review · Reviewer_Li4s · 2022-07-12

**Rating:** 6
**Confidence:** 4
**Soundness:** 3 good
**Presentation:** 3 good
**Contribution:** 2 fair

**Summary:**

In this paper, the problem of learning the optimal threshold policy for Markov Decision Processes (MDPs) is considered. Using the monotonicity property of threshold policies, the authors establish a simple policy gradient formula for the class of threshold policies. Using that, an off-policy actor-critic algorithm (DeepTOP) is proposed to learn the optimal policy in a situation where the optimal policy is known to possess a threshold structure. Moreover, the equivalence between obtaining the whittle index in a Restless Multi-armed Bandit (RMAB) problem and optimal threshold policy in an MDP is established. Following that, the DeepTOP algorithm is extended to the RMAB setting. Extensive simulation results are presented to demonstrate that the proposed algorithms outperform other algorithms in the literature.

**Questions:**

However, there are several concerns as stated below.
1.	The idea of viewing the Whittle index policy for RMABs as an optimal threshold policy is already developed in [1] as stated in the paper. Another important work in this direction is
[a] Robledo, F., Borkar, V., Ayesta, U., & Avrachenkov, K. (2022). QWI: Q-learning with Whittle Index. ACM SIGMETRICS Performance Evaluation Review, 49(2), 47-50.
See Equation (8) in the paper above.
How is the proposed algorithm in this paper different from the schemes described in these papers?
2.	It is assumed that $\lambda_t\in[-M,M]$ for all $t$ and the states can be numbered. This essentially translates into a finite state space. In the following paper, can’t Theorem 1 be derived as a corollary of the policy gradient theorem in
[b] Marbach, P., & Tsitsiklis, J. N. (2001). Simulation-based optimization of Markov reward processes. IEEE Transactions on Automatic Control, 46(2), 191-209.
3.	In the proof of Theorem 1, in the first step, the rationale behind swapping integration and summation is not clear. It needs to be explicitly stated in the paper.
4.	In Algorithm 1, why don’t the authors consider a decreasing $\epsilon$? Does constant $\epsilon$ guarantee convergence to the optimal solution?
5.	The deep threshold optimal policy computation of RMAB in Section 5 appears to be a straightforward extension of the policy gradient theorem in Theorem 1 because of the equivalence between obtaining the whittle index in a Restless Multi-armed Bandit (RMAB) problem and optimal threshold policy in an MDP. However, since this idea was already introduced in [1,a], the contribution in Section 5 is limited.
6.	The authors have performed extensive simulations on various problems such as electric vehicle charging problem, inventory management, and make-to-stock problem. By leveraging the monotone property, DeepTOP performs better than DDPG and TD3. However, the explanation regarding how it outperforms SALMUT is not clear.  Although DeepTOP employs the threshold policy gradient directly, if you take the policy gradient algorithm in [b] and encode the threshold policy information in the gradient of the transition probability matrix, is that not the same as the threshold policy gradient theorem (Theorem 1 in the paper)? The authors are requested to explain this point.


**Limitations:**

Overall, although the authors’ effort in exploiting the information regarding the existence of the threshold-based optimal policy in the learning framework is appreciable, the contribution regarding extension towards RMAB needs to be better highlighted. Moreover, how the policy gradient theorem (Theorem 1) presented in the paper is a non-trivial extension of the policy gradient theorem in [b] within the context of threshold policies, needs to be established clearly.

**Strengths And Weaknesses:**

The paper is well-written, and the claims appear to be correct. Extensive simulations have been performed to demonstrate the efficacy of the proposed approaches.

---

> ### Author Response · Authors · 2022-08-01
> **Response to reviewer Li4s comments - part 1**
>
>
> We thank the reviewer for the detailed comments. If our responses are satisfactory, we kindly ask the reviewer to update the score.
>
> ### *1. The idea of viewing the Whittle index policy for RMABs as an optimal threshold policy is already developed in [1] as stated in the paper. Another important work in this direction is [a] Robledo, F., Borkar, V., Ayesta, U., & Avrachenkov, K. (2022). QWI: Q-learning with Whittle Index. ACM SIGMETRICS Performance Evaluation Review, 49(2), 47-50. See Equation (8) in the paper above. How is the proposed algorithm in this paper different from the schemes described in these papers?*
>
> We thank the reviewer for the comment. We would like to emphasize that there are important differences between our DeepTOP and [1,a] in design principles, actual algorithms, theoretical properties, and simulation performance, which we detail below:
>
> **[Design principle]**
> The main design principle of [1,a] is that the Whittle index is the solution to the equation $Q(x,0) = Q(x,1)$ (Eq. (6) in [a]), instead of an optimal threshold policy. In fact, the word "threshold" does not appear in [a], and [1] specifically states that it does not learn the Whittle index as a threshold policy because it is hard to do so, "The Whittle index itself, however, is not a simple threshold, but a function of the state.... At the same time, Whittle index is defined in terms of an equality. So a much simpler scheme is used here, which makes incremental changes towards forcing this equality."
>
> In contrast, our DeepTOP views the Whittle index as the optimal threshold policy. We demonstrate that this view leads to a simple learning algorithm. Hence, our DeepTOP also solves the hard challenge described in [1].
>
> **[Algorithm design]**
> The algorithms in [1,a] are based on satisfying Eq. (6) in [a] for each individual state. As a result, the algorithms in [1,a] need to update the Whittle index of each state independently. This is evident in Eq. (8) in [a], which only concerns the state x.
>
> In contrast, DeepTOP aims to find the optimal threshold policy with respect to one single objective function, Eq. (8) in our paper. Hence, it only needs to apply Eq. (9) in our paper once.
>
> While the difference may seem subtle at first, it leads to significant difference in practice, as we will show below.
>
> **[Theoretical properties]**
> The algorithms in [1,a] are only applicable to indexable bandits due to their reliance on Eq. (6) in [a].
>
> In contrast, as we stated in the paper, our Theorem 2 makes no assumption on the indexability of bandits. Hence, DeepTOP can still be employed for bandits that are not indexable, and is guaranteed to find a locally optimal threshold policy for non-indexable bandits.
>
> **[Simulation performance]**
> We have implemented a neural-network extension of the algorithms in [1,a], which is called "Neural WIBQL" in Fig. 3 in our paper. Neural WIBQL uses the same neural network architecture and the same critic network update as our DeepTOP. The only difference is in the update of actor. In each update, Neural WIBQL updates the Whittle index of each state according to Eq. (8) in [a]. Our DeepTOP performs one single update according to Eq. (9) in our paper. As a result, Neural WIBQL is much slower than DeepTOP. For example, to run the setting in Fig. 3(a), Neural WIBQL takes 58 minutes and DeepTOP only takes 14 minutes.
>
> In addition, it can be seen that DeepTOP significantly outperforms Neural WIBQL. This shows that our algorithm is both more time-efficient and more sample-efficient than the algorithms in [1,a].

---

> > ### Author Response · Authors · 2022-08-01
> > **Response to reviewer Li4s comments - part 2**
> >
> > ### *2. It is assumed that  $\lambda_t \in [-M, M]$ for all $t$ and the states can be numbered. This essentially translates into a finite state space. In the following paper, can’t Theorem 1 be derived as a corollary of the policy gradient theorem in [b] Marbach, P., & Tsitsiklis, J. N. (2001). Simulation-based optimization of Markov reward processes. IEEE Transactions on Automatic Control, 46(2), 191-209.*
> >
> > First, we would like to emphasize that our state space is not finite. We assume that each state has two components, $\lambda$ and $v$. The component $v$ is from a finite and discrete set, and hence can be numbered. However, $\lambda$ can be any real number in $[-M, M]$. Hence, the state space is uncountably infinite.
> >
> > Second, the policy gradient theorem in [b] is not applicable to our paper. The paper [b] considers stochastic policies and employs the policy gradient theorem to update the probability distribution of actions. On the other hand, our paper considers threshold policies, which are deterministic policies as they will choose action = 1 with probability 1 if $\lambda$ is smaller than the threshold. Hence, [b] cannot be directly applied to threshold policies. One way to apply [b] for threshold policies is to approximate threshold policies by a stochastic policy. This is basically what SALMUT did. However, the approximation inevitably leads to inaccuracy. We will discuss the shortcomings of SALMUT in the response to comment 6.
> >
> > Since our Theorem 1 directly considers deterministic threshold policies, it cannot be derived from [b].
> > We will state this difference between theorem 1 and [b] in the updated paper.
> >
> >
> > ### *3. In the proof of Theorem 1, in the first step, the rationale behind swapping integration and summation is not clear. It needs to be explicitly stated in the paper.*
> >
> > We thank the reviewer for highlighting this step. We integrate over a finite sum of vector states $v$.
> > Since the $Q_{\mu^\phi}\Big(\lambda, v, 1(\mu^\phi(v) > \lambda)\Big)$ values are discounted, the finite sum converges for all Q-values.
> >
> > The integral $ \int_{\lambda = -M}^{\lambda = +M} \sum_{v\in\mathcal{V}}Q_{\mu^\phi}\Big(\lambda, v, 1(\mu^\phi(v) > \lambda)\Big) d\lambda <
> > \infty$
> > for the range $\lambda \in [-M, +M]$. Since the finite sum converges, using the Fubini-Tonelli theorem, the two terms are equal
> > $ \int_{\lambda = -M}^{\lambda = +M} \sum_{v\in\mathcal{V}}Q_{\mu^\phi}\Big(\lambda, v, 1(\mu^\phi(v) > \lambda)\Big) d\lambda = \sum_{v\in\mathcal{V}}Q_{\mu^\phi}\int_{\lambda = -M}^{\lambda = +M} \Big(\lambda, v, 1(\mu^\phi(v) > \lambda)\Big) d\lambda$.
> >
> > We added this description in theorem 1 proof in the uploaded rebuttal version.
> >
> > ### *4. In Algorithm 1, why don’t the authors consider a decreasing $\epsilon$? Does constant  guarantee convergence to the optimal solution?*
> >
> > We thank the reviewer for the suggestion. To respond to this comment, we trained DeepTOP-MDP (algorithm 1) and the baselines with a decaying $\epsilon$ at rate $1/500$ per timestep and an initial $\epsilon = 1$.
> > Results provided in figure 6 show that DeepTOP-MDP still outperforms other baselines with a decaying $\epsilon$.
> >
> > ### *5. The deep threshold optimal policy computation of RMAB in Section 5 appears to be a straightforward extension of the policy gradient theorem in Theorem 1 because of the equivalence between obtaining the whittle index in a Restless Multi-armed Bandit (RMAB) problem and optimal threshold policy in an MDP. However, since this idea was already introduced in [1,a], the contribution in Section 5 is limited.*
> >
> > We agree that the proof of Theorem 2 (for RMAB) is similar to that of Theorem 1 (for MDP), and we have stated as such in the paper. However, we would like to emphasize that RMAB is an important field of study. Hence, the applicability of DeepTOP for RMAB should be considered as a significant strength.
> >
> > As mentioned for question 1, there are important differences between [1,a] and our DeepTOP in multiple aspects.

---

> > > ### Author Response · Authors · 2022-08-01
> > > **Response to reviewer Li4s comments - part 3**
> > >
> > > ### *6. The authors have performed extensive simulations on various problems such as electric vehicle charging problem, inventory management, and make-to-stock problem. By leveraging the monotone property, DeepTOP performs better than DDPG and TD3. However, the explanation regarding how it outperforms SALMUT is not clear.*
> > >
> > >
> > > SALMUT has the following two important limitations:
> > >
> > > First, it requires the states to be pre-sorted by their indexes. In Section 3, [24] states "We consider the set of threshold policies where the thresholds for different events are ordered $(\tau(i) \geq \tau(j) \text{ for } i < j)$ and represent them as policies parametrized by the threshold vector $\boldsymbol{\tau} = [\tau(0), \tau(1), ..., \tau(N)]^T$ where $\tau(0) \geq \tau(1) \geq ... \geq \tau(N).$" In contrast, our DeepTOP does not require the knowledge of ordering.
> > >
> > > Second, SALMUT does not directly consider threshold policies, which are deterministic policies whose outcomes are not continuous. Instead, SALMUT approximates threshold policies by randomized policies based on sigmoid functions. (See Eq. (7) of [24]) SALMUT needs this approximation because it can only handle continuous and differentiable functions. We believe this approximation might be the reason why SALMUT is less accurate than DeepTOP.
> > >
> > > In contrast, DeepTOP directly considers deterministic threshold policies. In fact, the piece-wise constant behavior of threshold policies is the key part in the proof of Theorem 1. On line 115, we stated: "In other words, for any vector state $v$, the threshold policy would take the same action under all $\lambda \in (\mathbb{M}^{n+1}, \mathbb{M}^n)$, and we use $\pi^{n+1}(v)$ to denote this action."
> > >
> > >
> > > ### *7. Although DeepTOP employs the threshold policy gradient directly, if you take the policy gradient algorithm in [b] and encode the threshold policy information in the gradient of the transition probability matrix, is that not the same as the threshold policy gradient theorem (Theorem 1 in the paper)?*
> > >
> > > As explained in our response to question 2, this is not doable because threshold policies are deterministic policies. Also, the state space of our problem is not finite. Rather, it is uncountably infinite.

---

> > > > ### Comment · Reviewer_Li4s · 2022-08-09
> > > > **Thanks for the response**
> > > >
> > > > I thank the authors for their response. The response clarifies the paper better. I am satisfied with most of the authors’ responses. However, I am still not convinced about the contribution in the domain of RMAB as the analysis for MDPs and RMABs are similar with no major differences.
> > > > I have increased my review score accordingly.

---

### Meta-Review · Area_Chair_C7gu · 2022-08-24

**Recommendation:** Accept
**Confidence:** Certain

**Metareview:**

The paper considers a subset of dynamic problems, in which the optimal policy is a threshold-policy. The authors use this attribute to formulate tailored off-policy actor-critic algorithms, for both MDPs and RMABs which are gradient-based, so can utilize neural networks. They empirically compare their method to SOTA methods in three MDP domains and three RMAB parameterizations, the results show that their method, DeepTOP, performs better than the compared methods in all the experiments.

The paper is well written and the claims are correct.  The performance of DeepTOP compared to the other methods is impressive.  All four reviewers were on the positive side for acceptance.


**Award:**

No

---

### Decision · Program_Chairs · 2022-09-14

Accept